# Andrographolide Inhibits Corneal Fibroblast to Myofibroblast Differentiation *In Vitro*

**DOI:** 10.3390/biom12101447

**Published:** 2022-10-09

**Authors:** Vanessa Rozo, Melinda Quan, Theint Aung, Jennifer Kang, Sara M. Thomasy, Brian C. Leonard

**Affiliations:** 1Department of Surgical and Radiological Sciences, School of Veterinary Medicine, University of California, Davis, CA 95616, USA; 2Department of Ophthalmology & Vision Science, School of Medicine, University of California, Davis, CA 95616, USA

**Keywords:** andrographolide, corneal fibrosis, corneal wound healing, myofibroblast

## Abstract

Corneal opacification due to fibrosis is a leading cause of blindness worldwide. Fibrosis occurs from many causes including trauma, photorefractive surgery, microbial keratitis (infection of the cornea), and chemical burns, yet there is a paucity of therapeutics to prevent or treat corneal fibrosis. This study aimed to determine if andrographolide, a labdane diterpenoid found in *Andrographis paniculate*, has anti-fibrotic properties. Furthermore, we evaluated if andrographolide could prevent the differentiation of fibroblasts to myofibroblasts *in vitro*, given that the transforming growth factor beta-1(TGF-β1) stimulated persistence of myofibroblasts in the cornea is a primary component of fibrosis. We demonstrated that andrographolide inhibited the upregulation of alpha smooth muscle actin (αSMA) mRNA and protein in rabbit corneal fibroblasts (RCFs), thus, demonstrating a reduction in the transdifferentiation of myofibroblasts. Immunofluorescent staining of TGF-β1-stimulated RCFs confirmed a dose-dependent decrease in αSMA expression when treated with andrographolide. Additionally, andrographolide was well tolerated *in vivo* and had no impact on corneal epithelialization in a rat debridement model. These data support future studies investigating the use of andrographolide as an anti-fibrotic in corneal wound healing.

## 1. Introduction

Corneal fibrosis—the opacification of the outermost, transparent layer of the eye—represents one of the most common causes of vision loss globally. The World Health Organization estimates that corneal opacification accounts for over 4% of blindness worldwide, an estimated 1.5–2 million cases each year [1]. There are many causes of corneal fibrosis include ocular trauma (laceration/perforations and foreign bodies), photorefractive surgery, microbial keratitis, and acidic/alkaline agents (chemical burns). Topical corticosteroids reduce corneal inflammation and fibrosis, and mitomycin C is commonly utilized to prevent post-operative corneal fibrosis in patients undergoing keratorefractive therapy (e.g., LASIK surgery), but both treatments can cause vision-threatening complications [2,3,4,5,6]. Furthermore, there are no additional therapeutic strategies that directly intervene in the process of corneal fibrosis [1,7,8]. In severe instances of corneal fibrosis, corneal transplantation is needed to restore vision. Unfortunately, there is a supply shortage for donor corneas and this intervention is complicated by the risk of rejection [9]. Thus, there is a clinical need to develop targeted therapeutics that prevent corneal fibrosis in the context of chronic inflammation and infection, reducing the need for corneal transplantation all together [2,3,4].

During corneal wounding, the exposed stroma and their quiescent cells, termed keratocytes, are activated by inflammatory cytokines, TGF-β1 and IL-1α [10,11,12]. Activation of these keratocytes leads to their transdifferentiation into activated fibroblasts, and subsequently into myofibroblasts. This process is referred to as keratocyte-fibroblast-myofibroblast (KFM) transformation [12,13,14,15]. While KFM transformation is necessary for appropriate wound healing, the persistence of the myofibroblast phenotype, characterized by the presence of alpha smooth muscle actin (αSMA) expression, within the wound is associated with corneal fibrosis and stromal haze [12,16,17,18,19,20]. Despite an extensive literature base and an in-depth knowledge of the TGF-β1 signaling pathway, there are no effective therapeutics currently available to prevent or limit corneal fibrosis [21].

Andrographolide, a labdane diterpenoid found in *Andrographis paniculata*, has a plethora of anti-cancer, anti-bacterial, anti-viral, anti-inflammatory and anti-fibrotic properties. Specifically, the anti-fibrotic properties of andrographolide have been demonstrated in murine models of pulmonary and myocardial fibrosis [22,23]. The specific mechanism of action that leads to an anti-fibrotic response has not been fully elucidated; however, it is believed to be mediated through the suppression of the TGF-β1 signaling pathway, preventing Smad2/3 critical upstream regulators of αSMA expression [24,25].

Given that andrographolide has shown promise as an anti-fibrotic agent in murine models of fibrosis, the current study aimed to determine the effect of andrographolide in suppressing the transdifferentiation of rabbit corneal fibroblasts (RCFs) to myofibroblasts *in vitro*. Additionally, we sought to determine the maximum tolerated dose and effect on corneal epithelial wound healing in a rat model.

## 2. Materials and Methods

### 2.1. Primary Rabbit Stromal Cell Culture

Primary RCFs were isolated as previously described and used between passages 3 and 7 [26,27]. RCFs were cultured in Dulbecco’s modified medium (DMEM) low glucose with 10% fetal bovine serum (Atlanta Biologicals, Lawrence, GA, USA) and 1% penicillin-streptomycin-amphotericin B (PSF; Lonza, Walkersville, MD, USA).

### 2.2. Cell Viability Assays

MTT (3-[4,5-dimethylthiazol-2-yl]-2,5 diphenyl tetrazolium bromide) assays were carried out to assess the effects of various concentrations of andrographolide (Selleck Chemicals, Houston, TX, USA) on RCF cell viability using established methods in our lab [26]. The RCFs were plated at a density of 5000 cells per well in 100 μL of DMEM low glucose with 1% PSF (no serum) in a 96-well plate. The cells were incubated for 24 h to allow attachment to the bottom of the wells prior to treatment. At a density of 60–70% confluence, RCFs were treated for 24 h with varying concentrations of andrographolide ranging from 0.5 to 1000 μM in eight replicates [26]. Media and saponin (1 mg/mL) were used as negative and positive toxicity controls, respectively; dimethyl sulphoxide (DMSO) was used as a vehicle control [26]. Following treatment, 100 μL of MTT solution (0.5 mg/mL) was added to each well and incubated for 3 h at 37 °C. After the incubation period, the media was carefully removed from each well and 100 μL of DMSO; Sigma Chemical Co., St. Louis, MO, USA) was added to dissolve the formazan crystals. Absorbance values were measured at 540 nm with a microplate spectrophotometer (Synergy 4; BioTek, Instruments Inc., Winooski, VT, USA). The cell viability (%) was calculated relative to vehicle control wells: (absorbance of treated cells − absorbance of blank)/(absorbance of vehicle control − absorbance of blank) × 100.

### 2.3. Andrographolide Treatment

RCFs were plated in a 6-well plate at a density of 1 × 10^5^ cells per well. The following day, the media was changed to DMEM low glucose with 1% PSF and subsequently treated with andrographolide (25 μM or 75 μM) or vehicle control (DMSO), in the absence or presence of TGF-β1 (10 ng/mL). TGF-β1 at this concentration induces myofibroblast transformation in RCFs [25,28]. For mRNA/qPCR analysis, cells were harvested 24 h after treatment. For protein/Western blot analysis, cells were harvested 48 h after treatment due to the lag in protein translation compared with mRNA transcription, and to ensure adequate amounts of protein were isolated for analysis.

### 2.4. RNA Extraction and Quantitative Real-Time PCR

Total RNA was extracted from RCFs using the GeneJET RNA Purification Kit (Thermo Fisher Scientific, Waltham, MA, USA) following the manufacturer’s specifications. The RNA was quantified (NanoDrop ND-1000 spectrophotometer, Thermo Fisher Scientific). Quantitative real-time PCR was performed using equivalent amounts of RNA (varying 30–50 ng) with the SensiFAST Probe Hi-ROX One-Step kit (Bioline, Taunton, MA, USA) and TaqMan aptamers specific to rabbit glyceraldehyde 3-phosphate dehydrogenase (GAPDH, Oc03823402_g1; Thermo Fisher Scientific) or αSMA (ACTA2, Oc03399251_m1; Thermo Fisher Scientific) in the total volume of 10 μL per reaction, as previously described [26]. The GAPDH expression served as a housekeeping control. Gene expression data were calculated as previously reported [12] and normalized to the expression of mRNA from cells in the absence of both TGF-β1 and andrographolide. The qPCR was performed from three independent experiments.

### 2.5. Protein Extraction and Western Blot

Protein was extracted using a radioimmunoprecipitation assay (RIPA) buffer and Protease/Phosphatase Inhibitor Cocktail (Thermo Fisher Scientific). Each well was scraped and homogenized. Equivalent amounts of protein (varying from 2–10 μg) were loaded onto a 10% NuPAGE Bis-Tris gel (Life Technologies, Carlsbad, CA, USA). Gel electrophoresis was performed at 100 mA for 50 min per gel, followed by the transfer to a nitrocellulose membrane (Life Technologies) at 1.3 A for 10 min. The membrane was blocked at 37 °C for 1 h with 5% evaporated milk in tris buffered saline (TBS). The membrane was incubated with a mouse monoclonal primary antibody specific to anti-αSMA (A5228; Sigma-Aldrich, St. Louis, MO, USA) or anti-GAPDH (G8795; Sigma-Aldrich) diluted 1:3000 in tris buffered saline with 0.1% tween 20 (TBS-T) with 0.1% evaporated milk overnight at 4 °C. Blots were washed five times with TBS-T before incubating with peroxidase-conjugated goat anti-mouse antibody (SeraCare, MA 5450-0011) diluted 1:20,000 in TBS-T with 0.1% evaporated milk at room temperature for 1 h. The membrane was washed four more times with TBS-T and once with TBS and protein bands of interest were detected using a Western blotting detection kit (WesternBright Quantum, Advansta, Menlo Park, CA, USA). Densitometry analyses were carried out with ImageJ software (National Institutes of Health, Bethesda, MD, USA). The band densities of αSMA were normalized to GAPDH. Western blot analyses were performed from three independent experiments.

### 2.6. Immunofluorescence of RCFs

RCFs were cultured in a 12-well cell culture dish on glass coverslips and treated in the presence or absence of andrographolide (25 μM or 75 μM) and TGF-β1 (10 ng/mL) for 24 h as described above. The cells were washed with PBS, fixed with paraformaldehyde 4%, permeabilized in 0.1% triton, and blocked for 1 h at room temperature. Immunolabeling was performed with a mouse monoclonal primary antibody anti-αSMA (A5228; Sigma-Aldrich) at a 1:100 dilution in blocking buffer and allowed to incubate overnight at 4 °C. The following day, the coverslips were washed with phosphate-buffered saline with tween 20 (PBS-T). The secondary fluorescent goat anti-mouse IgG (H+L) Daylight 594 Conjugated (Invitrogen 35510) antibody and the AlexaFluor 488 phalloidin antibody (Invitrogen A12379) were added at a 1:200 dilution and incubated at room temperature for 1 h. The coverslips were then washed twice with PBS-T and then once with PBS. The nuclei were Hoescht 33342, trihydrochloride, trihydrate-stained (Invitrogen H3570 2 μg/mL), allowed to incubate at room temperature for 15 min, then washed with PBS. The coverslips were fixed to glass slides with Prolong Gold Antifade (Invitrogen P36930) reagent. Cells were imaged using the Keyence BZ-X810. Exposure times were kept the same amongst all samples to allow for comparison between samples.

### 2.7. Animals

Four Long Evans female rats (Charles River Laboratories, Wilmington, MA, USA) were used in this study with a mean ± SD bodyweight of 272 ± 31 g. The study was approved by the Institutional Animal Care and Use Committee of the University of California, Davis and performed in compliance with the Association of Research in Vision and Ophthalmology statement for the use of animals in vision research. A detailed ophthalmic examination was performed before inclusion in the study and all animals had no evidence of ophthalmic disease.

### 2.8. Maximum Tolerated Dose

A dose escalation study with four rats was performed to determine the maximum tolerated dose. On day 1, the highest effective dose based on the optimal *in vitro* dose was applied (one 10 μL drop andrographolide: 50 μM) to the right eye. The left eye was treated with the vehicle control, DMSO, at a concentration equivalent to that used in the right eye. Both eyes were assessed using the semiquantitative preclinical ocular toxicology scoring (SPOTS) system at the following timepoints: 15 and 30 min, 1, 4, and 24 h [29]. If no adverse events were recorded over the 24 h time period, the next highest dose the dose was administered and assessed as described. The dose escalation continued until an adverse event occurred, and the highest tolerable dose was used in subsequent experiments.

### 2.9. Debridement and Postoperative Care

After a 45-day washout, the same four rats were induced and maintained under anesthesia with isoflurane in a closed chamber with a surgical induction mask for the epithelial debridement procedure. The ocular surface was aseptically prepared with 0.2% povidone iodine and saline. The cornea of both eyes was anesthetized with 0.5% proparacaine hydrochloride ophthalmic solution (Alcon laboratories, Inc., Fort Worth, TX, USA) and the corneal epithelium was debrided using an excimer spatula (Beaver-Visitec, Waltham, MA, USA), such that there was only a thin rim of limbal epithelium circumferentially. The corneal epithelial debridement was confirmed by the retention of fluorescein sodium (NaFL) stain (BIO GLOTM; 1 mg, HUB Pharmaceuticals, LLC., Rancho Cucamonga, CA, USA) and subsequently imaged with a digital camera (Nikon D300, Nikon Co., Tokyo, Japan; flash 1/4, iso 250, F11; with cobalt blue filters [Blue-AWB, Nikon] flash coverings and a yellow lens filter [HMC 62 mm Y[K2], HOYA]) to establish the wound area at baseline. Atropine sulfate 1% (Akorn, Inc., Lake Forest, IL, USA) and ofloxacin 0.3% (Alcon, Hunengerg, Switzerland) ophthalmic solutions were administered to both eyes followed by artificial tear ointment (Rugby Laboratories, Inc., Duluth, GA, USA) to prevent corneal desiccation during anesthetic recovery. Buprenorphine (0.05 mg/kg, intramuscularly) was administered to provide analgesia. All animals were monitored every 10 min until they returned to their normal upright body positions. Post-debridement, rats were treated with ofloxacin 0.3% given four times daily, buprenorphine (0.07 mg/kg subcutaneously) twice a day and carprofen (5 mg/kg subcutaneously) given daily. The rats were treated with andrographolide (500 μM) in the right eye and DMSO (0.5%) in the left eye four times daily.

### 2.10. Ophthalmic Examination Scoring and Imaging

The rats were examined daily using a portable slit lamp (SL-17; Kowa Co. Ltd., Nagoya, Japan) and imaged for fluorescein staining twice daily (Nikon D300) to assess the epithelial wound healing area. The remaining wound area at each time point was measured using ImageJ software (version 1.52k), compared with the baseline wound area for each rat, and the percentage of the remaining wound area was calculated for each time point.

### 2.11. Statistical Analysis

Data were presented as mean ± standard deviation (SD) and statistical analyses were performed with GraphPad Prism 8 (GraphPad Software, San Diego, CA, USA). The MTT assay was analyzed with a one-way analysis of variance (ANOVA) with a Dunnett’s multiple comparisons test. All remaining data sets were compared with a two-way (ANOVA) with a Šidák’s multiple comparisons test. Significance was defined as *p* < 0.05 for all analyses.

## 3. Results

### 3.1. Andrographolide Had a Dose-Dependent Effect on Cell Viability

RCFs demonstrated a dose-dependent cytotoxicity using the MTT assay with increasing concentrations of andrographolide. Specifically, doses of andrographolide ≥121.5 μM resulted in significant reductions (*p* < 0.05) in MTT conversion to formazan, consistent with cell death (Figure 1). Based on these findings, low (25 μM) and high (75 μM) doses of andrographolide were selected for further *in vitro* investigations for their anti-fibrotic properties. These concentrations also formed the basis for *in vivo* dosing in the tolerability and wound healing experiments.

### 3.2. Andrographolide Prevented the Expression of αSMA in TGF-β1-Stimulated RCFs

Andrographolide-treated RCFs had a dose-dependent inhibition of the expression of αSMA mRNA and protein even when co-treated with TGF-β1 (Figure 2). Specifically, a ~120-fold decrease in mRNA expression and a 2.5-fold decrease in protein were observed. To parallel these findings, immunofluorescence of these cells was performed and a dose-dependent inhibition of αSMA signal was also observed. Phalloidin and Hoescht staining were used to highlight the morphology of the cell and nuclei (Figure 3). The cell morphology of our control, DMSO, appeared more myofibroblast-like with a flattened and expansile appearance with clear stress fibers. However, the cells treated with andrographolide appeared to have a more fibroblast-keratocyte phenotype with a more spindle-like, linear shape.

### 3.3. Tolerability and Effect of Andrographolide on Corneal Epithelial Wound Healing

Treatment with all doses (50–1000 μM) were relatively well-tolerated with mild blepharospasm observed with the highest dose (1000 μM). An epithelial debridement wound healing experiment was performed to assess the effect of andrographolide on corneal epithelial wound healing at the highest tolerated dose of andrographolide (500 μM). Following debridement, there were no differences in the time to re-epithelization between animals treated with andrographolide versus vehicle control (DMSO), with all animals being healed within 72 h of epithelial debridement (Figure 4).

## 4. Discussion

The current study determined that andrographolide can prevent the transdifferentiation of rabbit corneal fibroblasts into myofibroblasts *in vitro* in the presence of TGF-β1. Andrographolide inhibited the upregulation of αSMA mRNA and protein, and simultaneously inhibited the phenotypic transdifferentiation of fibroblasts to myofibroblasts. Additionally, andrographolide was well-tolerated and had no impact on corneal epithelialization in a rat debridement model. These data represent the foundation for investigating the use of andrographolide as an anti-fibrotic in the context of corneal wound healing.

Andrographolide has demonstrated anti-fibrotic effects in other organ systems [9,22]. Specifically, mice treated systemically with andrographolide had a reduction in αSMA mRNA and protein expression when compared with controls in a radiation-induced model of pulmonary fibrosis [22]. Furthermore, epithelial-mesenchymal transition, a key event for fibrosis, was not observed and there was also a reduction in pro-inflammatory cytokine secretion [22]. Additionally, in a myocardial injury model, treatment with systemic andrographolide showed suppression of inflammation, oxidative stress, apoptosis, cardiac fibrosis, and endothelial dysfunction [23]. In aggregate, these studies and ours provide the basis to examine the effect of andrographolide on corneal fibroblast to myofibroblast transdifferentiation *in vivo*.

While the current study did not investigate the mechanisms underlying the anti-fibrotic effect of andrographolide, previous studies have demonstrated its role in inhibiting TGF-β1 signaling. TGF-βR1 and TGF-βR2—cell surface receptors—are required to activate the pro-inflammatory cytokine, TGF-β1. This attachment results in the autophosphorylation and phosphorylation of SMAD transcription factors [30,31]. Smad2/3 proteins mediate the TGF-β1 pathway [31,32] and are homo-oligomers in the cytoplasm. When ligand activation occurs, TGF-βR1 kinase (ALK5) phosphorylates Smad2/3, creating the Smad4 complex. Smad4 translocates to the nucleus and regulates genes with SMAD binding elements associated with transcriptional co-activators or co-repressors [30,31,33,34,35,36]. In a previous bleomycin-induced pulmonary fibrosis model, NIH 3T3 fibroblasts and primary lung fibroblasts were treated with andrographolide and there was a reduction in TGF-β1-induced Smad2/3 phosphorylation in both cell types, suggesting its regulation by andrographolide decreases pulmonary fibrosis [24]. Future experiments will focus on defining the signaling pathways that lead to the anti-fibrotic effect of andrographolide in corneal wound healing.

To our knowledge, the tolerability and epithelial debridement experiments in the present study are the first to investigate the *in vivo* effects of topically applied andrographolide to the eye. The epithelium—specifically, the basement membrane— expresses TGF-β1 and other growth factors that cause KFM transformation [37]. Thus, a delay in epithelial wound healing with concomitant TGF-β1 secretion could lead to the persistence of myofibroblasts and stromal haze. Despite the small sample size in this preliminary study, the results from all four rats were very consistent. Future studies will include larger sample sizes based on power analyses using the current data. The observation that andrographolide does not impede re-epithelialization or cause stromal haze is a key initial step in its evaluation as a novel therapeutic for corneal fibrosis.

In conclusion, andrographolide-treated RCFs had decreased αSMA mRNA and protein expression and maintained a fibroblast phenotype in the presence of TGF-β1 *in vitro*. During the *in vivo* studies, corneal epithelial wound healing did not significantly differ between those treated with andrographolide versus vehicle control. In aggregate, these data suggest that andrographolide is a promising topical anti-fibrotic therapeutic, although additional studies are needed to assess its mechanism of action and *in vivo* efficacy to decrease corneal fibrosis.

## Figures and Tables

**Figure 1 biomolecules-12-01447-f001:**
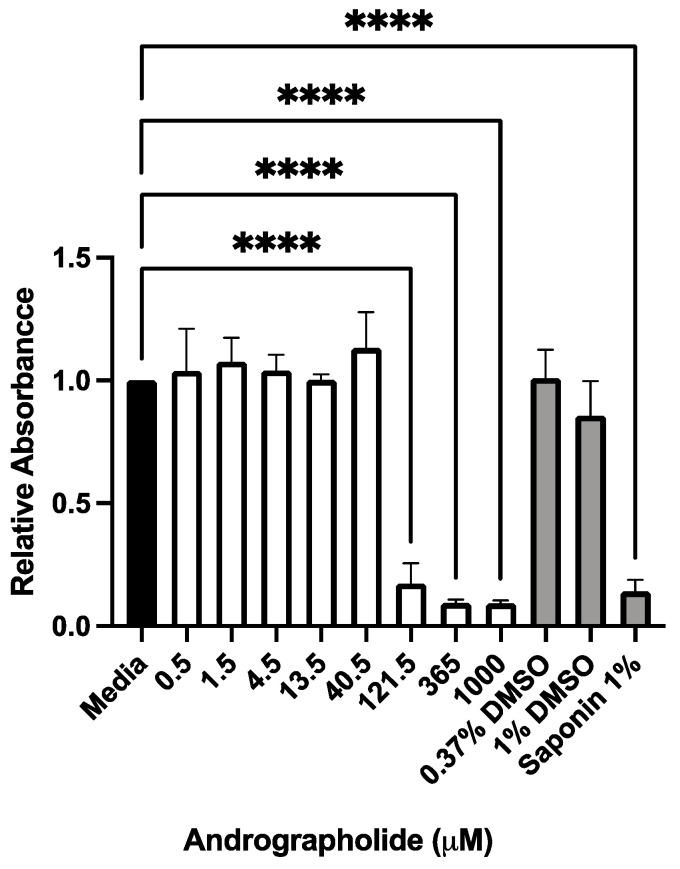
Andrographolide demonstrated a dose-dependent cytotoxicity in rabbit corneal fibroblasts (RCFs) *in vitro*. Treatment with andrographolide resulted in significant cytotoxicity in RCFs at concentrations of 121.5 μM and greater. Relative absorbance was normalized to media alone, DMSO treatments served as vehicle controls, saponin 1% served as a positive control for cytotoxicity. Bars indicate mean relative absorbance and error bars indicate standard deviation (*n* = 3 independent experiments). One-way ANOVA with a Dunnett’s multiple comparisons test performed with **** equivalent to *p <* 0.0001).

**Figure 2 biomolecules-12-01447-f002:**
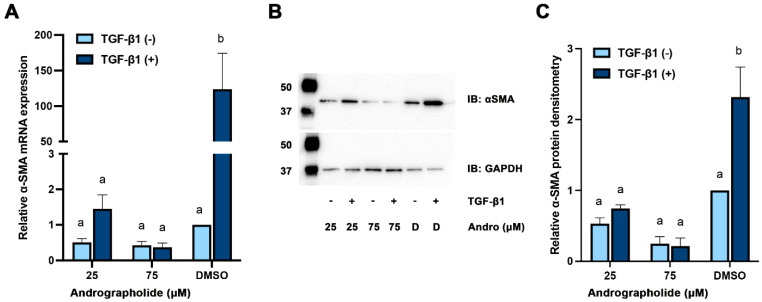
Treatment with andrographolide inhibits the TGF-β1-induced transdifferentiation of corneal fibroblasts into myofibroblasts. Primary rabbit corneal fibroblasts were treated with andrographolide (25 and 75 μM) or vehicle control (DMSO) in the presence or absence of TGF-β1 (10 ng/mL). (**A**) RT-qPCR was performed to measure the relative αSMA expression normalized to GAPDH. (**B**) Western blotting was performed with mouse monoclonal antibodies for αSMA or GAPDH, detected with a rabbit α-mouse HRP labelled secondary antibody and imaged using chemiluminescence. Ladder indicating 50 and 37 kDa. (**C**) Band densitometry was performed using ImageJ with normalizing αSMA to GAPDH, and further normalized to vehicle control without TGF-β1. Bars indicate mean expression and error bars indicate standard deviation (*n* = 3 independent experiments). Ladder present (kD). IB: immunoblot, Andro: Andrographolide, D: DMSO. Two-way ANOVA performed with a Šidák’s multiple comparisons test and different letters indicate statistically significant differences between groups ((**A**): a,b—*p =* 0.0001; (**C**): a,b—*p <* 0.005).

**Figure 3 biomolecules-12-01447-f003:**
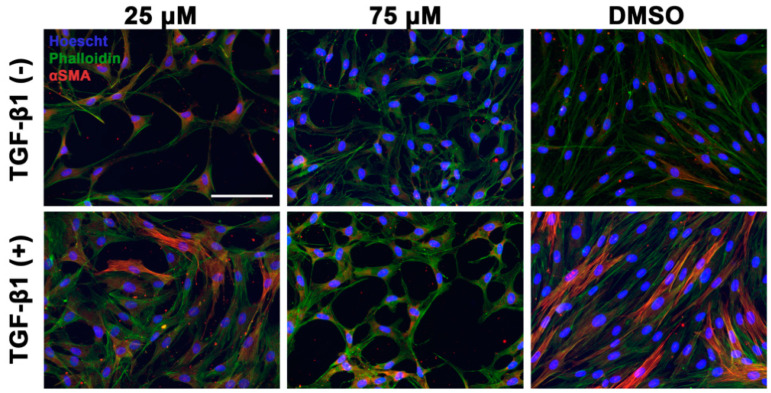
Immunofluorescence confirms a dose-dependent decrease in the upregulation of αSMA with immunofluorescent imaging. Primary rabbit corneal fibroblasts (RCFs) were treated with andrographolide (25 and 75 μM) or vehicle control (DMSO) in the presence or absence of TGF-β1 (10 ng/mL). RCFs treated with TGF-β1 resulted in increased stress fiber formation, F-actin (phalloidin), and αSMA expression. RCFs treated with andrographolide (25 μM and 75 μM) showed a decreased expression of αSMA with the 75 μM having a more pronounced effect. Images selected were representative of the overall immunofluorescence patterns associated with specific treatments. Hoescht nuclear stain in blue, phalloidin (F-actin) in green, αSMA in red. Scale bar equivalent to 100 μM.

**Figure 4 biomolecules-12-01447-f004:**
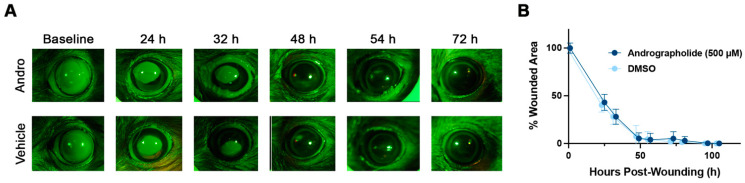
Treatment with topical ophthalmic andrographolide does not affect corneal epithelial wound healing after manual excimer spatula debridement of the corneal epithelium in rats. (**A**) Fluorescein-stained corneas demonstrating complete re-epithelialization of the cornea by 72 h in both andrographolide- and DMSO-treated Long Evans rats. Representative images of fluorescein-stained epithelial wounds in rats treated with andrographolide (500 μM) or vehicle control (DMSO 0.5%) four times daily. (**B**) There were no significant differences in epithelial wound healing rates in rat globes treated with andrographolide (500 μM, *n* = 4 eyes) or vehicle control (DMSO 0.5%, *n* = 4 eyes) (one way ANOVA with a Šidák’s multiple comparisons test, *p* > 0.05).

## Data Availability

Not applicable.

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
