# Peer review of "Andrographolide Inhibits Corneal Fibroblast to Myofibroblast Differentiation In Vitro"

_biomolecules, 2022, doi:10.3390/biom12101447_

Round 1

Reviewer 1 Report

This study showed the anti-fibrotic activity of andrographolide, a labdane diterpenoid, that can prevent the differentiation of fibroblasts to myofibroblasts in vitro. This activity was showed by inhibition of upregulation of mRNA and protein of alpha-SMA, and also, by immunofluorescence the authors showed a diminished expression of alpha-SMA in a dose dependent manner when cells were treated with andrographolide. The anti-fibrotic activity was also analyzed in vivo, and the results showed that had no effect on corneal epithelialization in a rat debridement model. The authors based on these results suggest that andrographolide could be used as a anti-fibrotic therapeutic, however they also establish that more studies must be asses in order to determine the mechanism of action of this compound. 

However, some points must be clarified:

1. The authors used two concentrations of andrographolide (25 and 75 uM), even authors mentioned that concentrations over 121.5 uM showed significant reductions, however, the reason for use that concentrations is not well explained in the text.

2. In figure 2C, bars corresponding to relative alpha-SMA protein densitometry in 25 uM andrographolide concentrations, not correspond with the bands in Figure 2B in that concentration. 

Author Response

Thank you all for your thoughtful and timely reviews. We have made the requested corrections and believe that the manuscript is stronger as a result.

REVIEWER #1

  1. The authors used two concentrations of andrographolide (25 and 75 uM), even authors mentioned that concentrations over 121.5 uM showed significant reductions, however, the reason for use those concentrations is not well explained in the text.

We chose to investigate the 25 and 75 uM since the MTT assay demonstrated significant cell death at concentrations ≥121.5 uM. By contrast, treatments with 25 and 75 uM did not affect cell viability and were thus selected for subsequent investigation.

We have tried to highlight this finding in the Results section with the following line, “Specifically, doses of andrographolide ≥ 121.5 uM resulted in significant reductions (P < 0.05) in MTT conversion to formazan, consistent with cell death (Figure 1). Based on these findings, low (25 uM) and high (75 uM) doses of andrographolide were selected for further in vitro investigation for their anti-fibrotic properties. Page 5, Line 213, Page 6, Lines: 214-216.

  1. In figure 2C, bars corresponding to relative alpha-SMA protein densitometry in 25 uM andrographolide concentrations, not correspond with the bands in Figure 2B in that concentration. 

Thank you for the comment. We have replaced the image with one that better represents the overall results. The densitometry figure (Figure 2C) represents the mean +/- SD of three experiments. Page 7, Line: 238.

Author Response

REVIEWER #2

  1. My only concern is related to the in vivo experiment, as the number of the eyes taken into consideration (n=4) might not be relevant to support the results. Usually, a number n>10 is necessary for relevant outcomes. The authors may refer to a recent work describing an animal model (https://doi.org/10.3390/ma15093374) in which ImageJ-processed photographs of eyes from different groups was performed in order to get better interpretation. However, the authors should mention the limitations of their study at the end of the Discussion section.

Thank you for this comment. We have added two sentences in the discussion detailing this limitation. We will base our sample sizes of future experiments on power analyses from data in the current study. We have added the following sentences, “Despite the small sample size in this preliminary study, the results from all four rats demonstrated little variability. Future studies will include larger sample sizes based on power analyses conducted with the current data.” Page 9, line 318-320.

Reviewer 3 Report

The authors have been studying the inhibition of keratocyte-fibroblast-myofibroblast transformation by andrographolide on primary rabbit corneal fibroblasts in vitro, dose escalation studies of andrographolide in Long Evans female rats, and corneal epithelial debridement study in vivo. Previous studies have demonstrated that andrographolide induces cell cycle arrest, inhibit proliferation in different cell types and block TGF-b1/Smad2 pathway. Therefore, there is an interesting idea of evaluating how the differentiation of corneal stomal fibroblasts to myofibroblasts could be modulated along the differentiation process when in contact of traditional medicine found in Andrographis paniculata plant.

The introduction is systemically performed, and Material and Methods are nicely written.

The overall impression along the manuscript is that despite experimental methodology are sound, however, the manuscript looks still as a preliminary study.

Here is the list of points the authors should consider, and I believe it would improve their work.

Requested revisions:

In methodology:

Line 93-94: Gene expression was measured 24-hours after treatment. Why the protein/western blot analysis was done 48-hours after treatment? What are consequences to primary gene and protein expressions? Please justify the different treatment times.

Lines 75, 89: Cell viability assays (MTT) and andrographolide treatments for qPCR and western blot assays were performed at different cell densities (higher and lower, cells/cm2), what are consequences to primary cell viability, morphology, and protein expression?

Lines 192-193: Statistical analysis: Include information on multiple comparisons tests that there performed for one-way ANOVA and two-way ANOVA. Include the information also in figure legends 1, 2, 4.

Figure legends need to define the statistical significances both ** and *** (Figure 1) and “a” and “b” (Figure 2). These are missing.

Line 145: Four Long Evans female rat were included in the in vivo studies. It is not clear how many animals were used for the 2.8. Maximum Tolerated Dose and 2.9. Debridement studies. If all four rats were used for both studies, then the washout period (dependent on the pharmacokinetic profile of andrographolide) must be adequate and should be presented and justified in the manuscript.

Line 204: Authors show in the Figure 1, a trend towards increased cell viability/increased cell proliferation at concentrations 0.5-13.5 uM, and statistically significant (P < 0.01 **) increase in cell viability/increased cell proliferation at 40.5 uM andrographolide. In addition, this is somehow controversial to Figure 3, where vehicle control (DMSO) images show remarkably higher density of rabbit corneal fibroblast than in 25 uM or 75 uM andrographolide treated cells. The images suggest that andrographolide inhibit cell proliferation. However, it is not supported by the MTT test presented in Figure 1. This needs to be discussed thoroughly.

Minor revisions:

line 12 and line 52: Andrographis paniculata, not paniculate

line 14, 16 and 17: open the abbreviations TGF-b1, aSMA, RCF in the abstract

line 68: is the abbreviation “FBS” needed if used only once?

line 80: perhaps DMSO should be opened already, not in line 83

line: 173: use the same font for “ofloxacin”

line 176, 179, 180: standardize the wording (now mixed styles for administration routes: IM, subcutaneously, and SQ)

line 215: typo in “immunofluorescence”

line 222: ladders labeled with band sized (kDa) should be included in the Figure 2B. In addition, the GAPDH loading control is highly inconsistent in the selected representative (?) image, please consider changing the image, e.g. “2022.4.15 WB RCFs treated with Andro (GAPDH, P7.2)”, if equivalent blot.

Author Response

REVIEWER #3:

  1. In methodology: Line 93-94: Gene expression was measured 24-hours after treatment. Why the protein/western blot analysis was done 48-hours after treatment? What are consequences to primary gene and protein expressions? Please justify the different treatment times.

Protein analysis was performed at 48 hours to ensure adequate amounts of protein were synthesized by the cells for western blot analysis. It is possible that the levels of aSMA upregulation (both mRNA and protein) could be variable depending on the duration of induction with TGF-B1, however the time points selected are within the typical ranges for analyzing this upregulation of aSMA (1-5).

We adjusted the sentence to, “For protein/western blot analysis, cells were harvested 48-hours after treatment due to the lag in protein translation compared with mRNA transcription, and to ensure adequate amounts of protein were isolated for analysis.” Page 3, Lines: 103-105.

  1. Lines 75, 89: Cell viability assays (MTT) and andrographolide treatments for qPCR and western blot assays were performed at different cell densities (higher and lower, cells/cm2), what are consequences to primary cell viability, morphology, and protein expression?

The MTT assay was seeded at a higher density (15,625 cells/cm2) when compared to the 6-well plate (10416 cells/cm2). The resulting confluence of the cells were roughly equivalent (60-70%) in both the 96-well plate MTT assay and 6-well treatment assay at the time of treatment. Additionally, we performed additional experiments to indicate no loss in cell viability after treatment in 6-well plates (data not shown). We did not appreciate any effects on cell morphology other than the stated changes seen with immunofluorescence based on expect treatment effects (fibroblast vs. myofibroblast phenotype).

  1. Lines 192-193: Statistical analysis: Include information on multiple comparisons tests that there performed for one-way ANOVA and two-way ANOVA. Include the information also in figure legends 1, 2, 4.

We have included the details of the multiple comparisons tests in both the Materials and Methods and the Figure Legends. Page 5, Lines: 204-206; Page 6, Lines: 223-224; Page 7, Lines: 247-249; Page 8, Lines: 275-277.

  1. Figure legends need to define the statistical significances both ** and *** (Figure 1) and “a” and “b” (Figure 2). These are missing.

We have added **** for Figure 1 and defined the significance in the legend. We have kept the a-b nomenclature in Figure 2 and listed the significance of a-b as P = 0.0001 in Figure 2A. We also listed the significance of a-b as P < 0.005 in Figure 2C as all comparisons were below P of 0.005. Page 6, Lines: 223-224; Page 7, Lines: 246-248.

  1. Line 145: Four Long Evans female rat were included in the in vivo studies. It is not clear how many animals were used for the 2.8. Maximum Tolerated Dose and 2.9. Debridement studies. If all four rats were used for both studies, then the washout period (dependent on the pharmacokinetic profile of andrographolide) must be adequate and should be presented and justified in the manuscript.

The same four rats were used in all studies. There was a 45-day washout between the Maximum tolerated dose study and the debridement study. There are no known ocular pharmacokinetic profiles for andrographolide, however, we would expect a 45-day washout would allow complete elimination of andrographolide.

We have clarified this in the text with the following sentence, “After a 45-day washout, the same four rats were induced and maintained under anesthesia with isoflurane in a closed chamber with a surgical induction mask for the epithelial debridement procedure.” Page 5, Lines: 175-177.

  1. Line 204: Authors show in the Figure 1, a trend towards increased cell viability/increased cell proliferation at concentrations 0.5-13.5 uM, and statistically significant (P < 0.01 **) increase in cell viability/increased cell proliferation at 40.5 uM andrographolide. In addition, this is somehow controversial to Figure 3, where vehicle control (DMSO) images show remarkably higher density of rabbit corneal fibroblast than in 25 uM or 75 uM andrographolide treated cells. The images suggest that andrographolide inhibit cell proliferation. However, it is not supported by the MTT test presented in Figure 1. This needs to be discussed thoroughly.

Despite an equal seeding density, there were areas of higher density on the coverslips. Figure 3 does demonstrate differences in density, however, the selected images reflect the overall changes in immunofluorescence intensity of the entire field. Repeat statistical analysis of the lower concentrations revealed nonsignificant increases in cell viability/increased proliferation. We apologize for this confusion.

We included the following sentence was added to highlight the overall immunofluorescence patterns in the selected images: Images selected were representative of the overall immunofluorescence patterns associated with specific treatments.” Page 7, line 256-257.

Minor revisions:

  1. line 12 and line 52: Andrographis paniculata, notpaniculate

This has been corrected. Page 1, Line: 15; Page 2, Line: 58.

  1. line 14, 16 and 17: open the abbreviations TGF-b1, aSMA, RCF in the abstract

This has been corrected. Page 1, Lines: 16-19.

  1. line 68: is the abbreviation “FBS” needed if used only once?

This abbreviation has been deleted.

  1. line 80: perhaps DMSO should be opened already, not in line 83

This was edited where DMSO was first mentioned. Page 3, Line 89-90.

  1. line: 173: use the same font for “ofloxacin”

This has been edited. Page 5, Line 186.

  1. line 176, 179, 180: standardize the wording (now mixed styles for administration routes: IM, subcutaneously, and SQ)

This has been addressed. Page 5, Lines 190-192.

  1. line 215: typo in “immunofluorescence”

This has been addressed. Page 6, Line 231.

  1. line 222: ladders labeled with band sized (kDa) should be included in the Figure 2B. In addition, the GAPDH loading control is highly inconsistent in the selected representative (?) image, please consider changing the image, e.g. “2022.4.15 WB RCFs treated with Andro (GAPDH, P7.2)”, if equivalent blot.

Thank you for noting this inconsistency. We have replaced Figure 2B with another western blot including a chemiluminescent ladder. Page 7, Lines 244-245; Figure 2B.

Round 2

Reviewer 3 Report

The manuscript has been improved and is now acceptable for publication.